# The impact of hyperglycemia on prognosis in chronic obstructive pulmonary disease patients with coronary heart disease: A five-year prospective study

Zhongshang Dai[1☯], Chenjie He[2☯], Huiui Zeng[2,3,4,5], Yan Chen ![ORCID][2,3,4,5]*

**1** Department of Infectious Diseases, The Second Xiangya Hospital, Central South University, Changsha, Hunan, People's Republic of China, **2** Department of Respiratory and Critical Care Medicine, the Second Xiangya Hospital, Central South University, Changsha, Hunan, China, **3** Research Unit of Respiratory Disease, Central South University, Changsha, Hunan, China, **4** Clinical Medical Research Center for Respiratory and Critical Care Medicine, Changsha, Hunan, China, **5** Diagnosis and Treatment Center of Respiratory Disease, Central South University, Changsha, Hunan, China

☯ These authors contributed equally to this work.
* chenyan99727@csu.edu.cn

## Abstract

### Background

There is a lack of research on the impact of hyperglycemia on the prognosis of patients with both chronic obstructive pulmonary disease (COPD) and coronary heart disease (CHD). This study aims to investigate the effect of hyperglycemia on long-term acute exacerbation frequency and mortality rates in COPD patients with CHD.

### Methods

This real-world and prospective study recruited inpatients with COPD from the second Xiangya Hospital of Central South University, in China in December 2016, with follow-up until March 2023. Moreover, we collected data on COPD participants from the National Health and Nutrition Examination Survey (NHANES), spanning the period from 1999 to 2018. All patients included in the study had concurrent CHD and available fasting blood glucose (FBG) measurements at the time of admission. Patients were categorized into normal blood glucose and hyperglycemia groups based on whether their FBG exceeded 7 mmol/L. Self-administered questionnaires, clinical records, and self-reported data were the primary methods for data collection. Kaplan–Meier survival analyses and Cox proportional hazard models were used to assess the risk of acute exacerbation and all-cause mortality for COPD patients with CHD during the follow-up period.

**Data availability statement:** All data of this study are available from the the Ethics Committee of the Second Xiangya Hospital of Central South University(contact via LiZhuo75@csu.edu.cn) for reasonable request.

**Funding:** This work was supported by the National Natural Science Foundation of China (No. 82370054 and 82070049), the Natural Science Foundation of Hunan Province (No. 2024JJ6560) and the National Key Clinical Specialty Construction Projects of China.

**Competing interests:** The authors have declared that no competing interests exist.

## Results

Among patients with both COPD and CHD, patients in the hyperglycemia group had lower smoking index, higher prevalence of diabetes, lower eosinophil percentage (Eos%), notably elevated C-reactive protein (CRP) and brain natriuretic peptide (BNP) levels, reduced albumin (ALB) levels, and higher incidence of fungal positivity than patients in the normal blood glucose group (p < 0.05). Age (HR = 1.098, 95% CI = 1.013–1.191, p = 0.024), hyperglycemia (HR = 3.622, 95% CI = 1.08–12.15, p = 0.037), and comorbidities with obsolete pulmonary tuberculosis (HR = 3.185, 95% CI = 1.03–9.85, p = 0.044) were identified as independent predictors of mortality in multivariate analyses over the follow-up years. Hyperglycemia, age, smoking, white blood cell count (WBC), uric acid (UA), creatinine (Cr), ALB, and aspartate aminotransferase (AST) were also identified as independent predictors of mortality in multivariate analyses during the follow-up years, according to NHANES data. Kaplan–Meier survival curves demonstrated that patients in the hyperglycemia group experienced significantly higher rates of exacerbations and mortality compared to those in the normal blood glucose group over a follow-up period of 5 years or more (log-rank test, p < 0.05).

## Conclusions

Hyperglycemia serves as an independent risk factor for prolonged acute exacerbations and mortality in COPD patients complicated with CHD post-discharge. Proactive intervention strategies targeting hyperglycemia should be promptly instituted to mitigate the risk of future acute exacerbations and mortality in COPD patients with CHD.

## Introduction

Chronic obstructive pulmonary disease (COPD) is a heterogeneous disease characterized by progressive and partially reversible airflow limitation, airway inflammation, and respiratory symptoms [1]. According to the 2019 Global Burden of Disease report, COPD affected 212 million individuals globally, resulting in approximately 3.28 million deaths [2]. COPD frequently coexists with other diseases, significantly impacting prognosis. Cardiovascular disease is a prevalent and critical complication of COPD [3]. The China Pulmonary Health Study reported a COPD prevalence of 8.6% among adults over 20 years old, with a higher prevalence in males than females [4].

Coronary heart disease (CHD) commonly coexists with COPD [5], and individuals with COPD are at an elevated risk of developing CHD. Additionally, CHD significantly increases the risk of readmission and mortality in patients with COPD [6]. COPD and CHD share common pathophysiological mechanisms and risk factors, including smoking, aging, and chronic systemic inflammation, leading to a high prevalence of comorbidity in elderly populations (20–30%). This coexistence is not merely additive but synergistic: acute COPD exacerbations may precipitate myocardial ischemia through hypoxemia, sympathetic activation, and increased ventricular afterload,

whereas impaired cardiac output in CHD can exacerbate ventilatory-perfusion mismatch, worsening COPD-associated hypoxia and hypercapnia [7,8]. Hyperglycemia is a prevalent comorbidity in patients with either COPD or CHD, with an incidence rate ranging from 36% to 80% among those with COPD [9,10]. Patients with COPD and concurrent hyperglycemia have prolonged hospital stays, an increased risk of readmission, and higher mortality rates [11–13]. Similarly, patients with CHD and concurrent hyperglycemia face a significantly elevated risk of adverse cardiovascular and cerebrovascular events, and increased long-term mortality [14,15]. Hyperglycemia is associated with a poor prognosis in individuals with either COPD or CHD. However, there is a lack of research on the impact of hyperglycemia on the prognosis of patients with both COPD and CHD. Hence, in this study, we aimed to investigate the effect of hyperglycemia on long-term acute exacerbation frequency and mortality rates in COPD patients with CHD.

## Methods

### Study population

This study received approval from the Ethics Committee of the Second Xiangya Hospital of Central South University (No. 2016076fabh001) and was carried out in compliance with the Declaration of Helsinki. Written informed consent was obtained from all participants.

This real-world, observational, prospective study enrolled inpatients with COPD at the Second Xiangya Hospital of Central South University between December 2016 and March 2023. Recruitment for the study commenced in December 30, 2016, with follow-up until March 5, 2023. Patient hospitalization data were retrieved from the medical record system on 05/06/2023. All participants were diagnosed with COPD in accordance with the Global Initiative for Chronic Obstructive Lung Disease 2020 guidelines [16]. Additionally, all patients included in the study had concurrent CHD and available fasting blood glucose (FBG) measurements at the time of admission. The study sample was confined to patients hospitalized due to COPD, excluding those with tumors, severe hepatic or renal diseases, severe heart failure, or severe complications of diabetes (e.g., ketoacidosis or hyperosmolar hyperglycemic syndrome). Furthermore, patients who had used systemic corticosteroids within three months prior to admission were excluded. Moreover, we collected data on COPD participants from the National Health and Nutrition Examination Survey (NHANES), spanning the period from 1999 to 2018. We specifically included participants with a history of CHD, excluding those without available FBG data. Patients were categorized into normal blood glucose and hyperglycemia groups based on whether their FBG exceeded 7 mmol/L.

### Study design

Patient hospitalization data were obtained from the medical record system of the Second Xiangya Hospital of Central South University, and follow-up information was gathered via telephone interviews. Data collection was conducted by trained research assistants, while variable verification was performed by trained physicians. Patient hospitalization data were retrieved from the medical record system on 05/06/2023.

The medical record system data encompass general demographic information (including age, gender, body mass index (BMI), smoking history, comorbidities, COPD Assessment Test (CAT) score, modified Medical Research Council (mMRC) dyspnea grade, 6-minute walk test (6MWT), and the number of COPD exacerbations in the preceding year), clinical laboratory data (such as fasting blood glucose (FBG) at admission, lung function, complete blood count, inflammatory markers, arterial blood gases, liver and kidney function, brain natriuretic peptide (BNP), and pathogen data), and current treatment information (including antibiotic treatment course, hormone treatment course, oxygen therapy, and invasive ventilation). Self-administered questionnaires, clinical records, and self-reported data were the primary methods for data collection. All pulmonary function tests were administered by laboratory technicians at the pulmonary clinics. Exacerbation was defined as an acute worsening of respiratory symptoms requiring a short-acting bronchodilator (SABD) only (mild), SABD plus antibiotics and/or oral corticosteroids (moderate), or hospitalization/emergency department visits (severe) [16].

All patients were followed up for a period exceeding 12 months. We collected data on acute exacerbations of AECOPD during the first, third, and fifth years of follow-up, as well as all-cause mortality data throughout the entire follow-up period. The specific follow-up process was conducted by trained research assistants through telephone consultations or outpatient interviews with patients and their relatives (at least once per year). Data on annual exacerbations and survival status, along with outpatient and inpatient medical records provided by the patients, were uniformly recorded in a database, with data collected every three months. The primary outcomes included hospitalization due to COPD exacerbation and overall mortality. Additionally, data were collected on age, gender, BMI, smoking history, comorbidities, lung function, complete blood count, liver and kidney function, and follow-up data from NHANES.

## Statistical analysis

Data processing was performed using SPSS version 21.0. The overall dataset did not follow a normal distribution. Categorical variables were compared using chi-square tests, whereas continuous variables were analyzed with Mann-Whitney U tests. Categorical data are represented by frequencies and percentages, while continuous data are reported as medians with interquartile ranges (25th and 75th percentiles). Kaplan-Meier survival analysis and Cox regression analysis were utilized to assess the impact of hyperglycemia on long-term mortality in COPD patients with CHD. Statistical significance was defined as $p < 0.05$.

## Results

The study's flowchart is illustrated in Fig 1. A total of 148 patients diagnosed with both COPD and CHD were recruited for the study from the Second Xiangya Hospital of Central South University, of which 47 were categorized under the

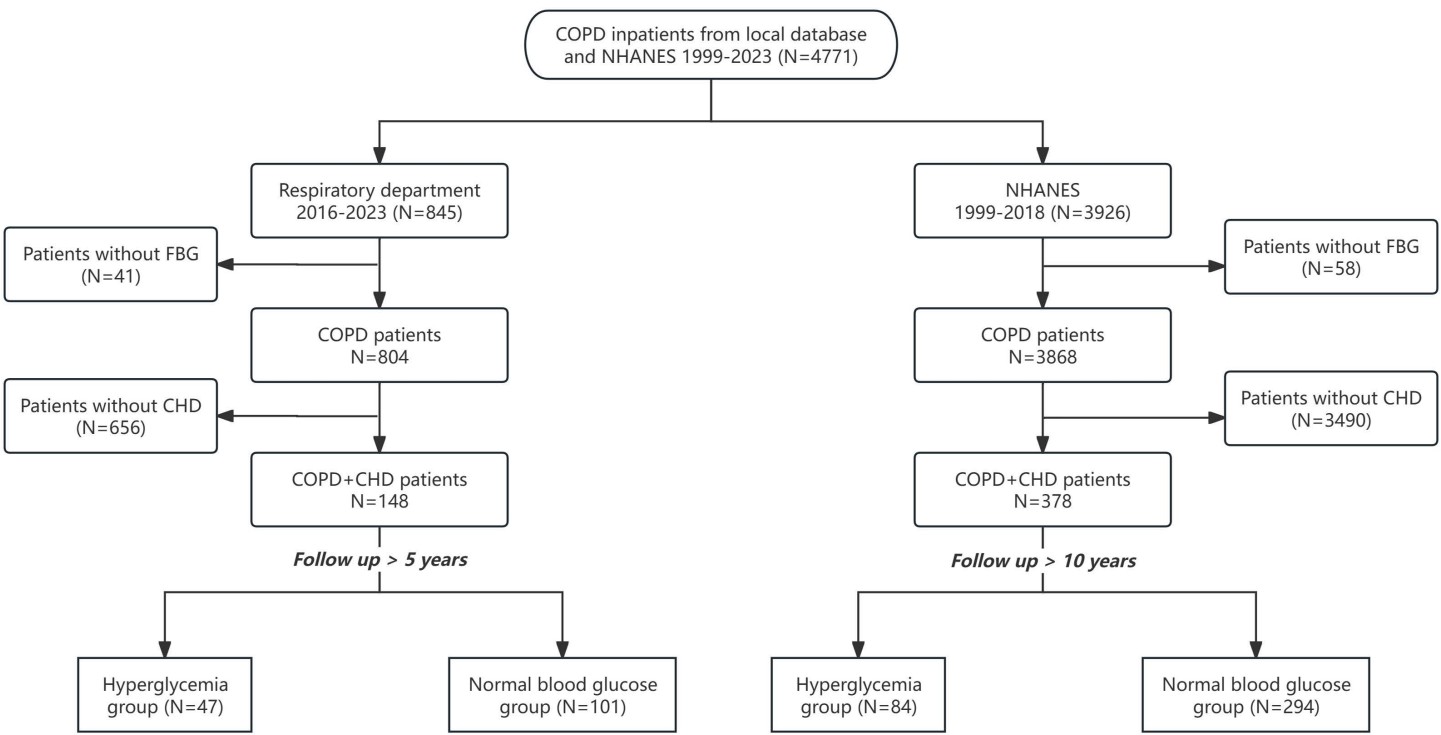

**Fig 1. Flow diagram.** COPD: Chronic obstructive pulmonary disease, NHANES: National Health and Nutrition Examination Survey, FBG: fasting blood glucose, CHD: coronary heart disease.

hyperglycemia group. Additionally, data derived from the NHANES comprised a cohort of 378 patients with concurrent COPD and CHD, among whom 84 were classified within the hyperglycemia group.

When comparing the baseline characteristics between the two cohorts, the smoking index in the hyperglycemia group was notably lower than that observed in the normal glucose group (p = 0.030). However, no statistically significant differences were observed between the two cohorts in terms of age, gender, BMI, smoking status, comorbid conditions such as pneumonia, asthma, bronchiectasis, prior pulmonary tuberculosis, hypertension, obstructive sleep apnea (OSA), and pulmonary heart disease, as well as 6MWT, CAT, mMRC scores at admission, and the frequency of acute exacerbations in the preceding year (Table 1).

Among patients with both COPD and CHD, the eosinophil percentage (Eos%) in the hyperglycemia group was significantly lower compared to the non-hyperglycemia group (p = 0.031), while the C-reactive protein (CRP) levels were notably elevated in the hyperglycemia group relative to the normal blood glucose group (p = 0.034). The incidence of fungal positivity, as indicated by sputum smear and culture, was higher in the hyperglycemia group than in the normal blood glucose group (p = 0.045). Compared to the normal blood glucose group, patients in the hyperglycemia group exhibited significantly elevated BNP levels (p = 0.036) and reduced albumin levels (p = 0.002). Additionally, the FBG levels in the hyperglycemia group were significantly higher than those in the normal blood glucose group (p = 0.012). No statistically significant differences were identified in lung function, white blood cell count (WBC), lymphocyte-to-neutrophil ratio (LNR), eosinophils, erythrocyte sedimentation rate (ESR), procalcitonin (PCT), interleukin-6 (IL-6), partial pressures of carbon dioxide (PaCO2) and oxygen (PaO2), uric acid (UA), creatinine (Cr), as well as viral and bacterial markers (Table 2).

**Table 1. Comparison of baseline data between the normal blood glucose group and the hyperglycemia group[*].**

| Items | Normal blood glucose group (N = 101) | Hyperglycemia group (N = 47) | P value |
|---|---|---|---|
| **Demographic characteristics** | | | |
| Age (years) | 73 (69,78) | 74 (67,80) | 0.481 |
| Male, n (%) | 87 (86.14%) | 37 (78.72%) | 0.255 |
| BMI (kg/m$^2$) | 21.41 (18.9,24.34) | 22.33 (19.61,24.89) | 0.316 |
| *Smoking status, n (%)* | | | 0.229 |
| Non-smoking | 19 (19%) | 14 (29.79%) | |
| Smoking | 32 (32%) | 10 (21.28%) | |
| Stopped smoking | 49 (49%) | 23 (48.94%) | |
| Smoking pack-years | 40 (15,60) | 30 (0,40) | 0.030 |
| *Comorbidities, n (%)* | | | |
| Pneumonia | 7 (6.93%) | 4 (8.51%) | 0.743 |
| Asthma | 10 (9.9%) | 5 (10.64%) | 0.890 |
| Bronchiectasia | 10 (9.9%) | 6 (12.77%) | 0.601 |
| Old pulmonary tuberculosis | 28 (27.72%) | 14 (29.79%) | 0.795 |
| OSA | 2 (1.98%) | 3 (6.38%) | 0.168 |
| Pulmonary heart disease | 19 (18.81%) | 9 (19.15%) | 0.961 |
| Hypertension | 55 (54.46%) | 29 (61.7%) | 0.407 |
| Admission 6MWT(m) | 281 (120,360) | 204 (10,344) | 0.210 |
| Admission mMRC | 3 (2,4) | 3 (2,4) | 0.674 |
| Admission CAT | 23 (18,28) | 25 (18,31) | 0.380 |
| Acute exacerbations in the previous year | 2 (1,3) | 1 (1,2) | 0.107 |

[*]Presented as MD (IQR) unless specified otherwise.

BMI: Body mass index, OSA:obstructive sleep apnea, 6MWT: 6-minute walk test, CAT: COPD Assessment Test, mMRC: Modified Medical Research Council, IQR: interquartile range, MD: median.

**Table 2. Comparison of laboratory data between the normal blood glucose group and the hyperglycemia group**[*].

| Items | Normal blood glucose group (N = 101) | Hyperglycemia group (N = 47) | P value |
|---|---|---|---|
| **Laboratory data** | | | |
| FEV1 (L) | 0.82 (0.62,1.08) | 0.81 (0.58,0.98) | 0.384 |
| FEV1(%) | 35.4 (27.15,47.2) | 37 (30.75,49.4) | 0.504 |
| FEV1/FVC (%) | 40.49 (32.7,53.9) | 43 (32.6,56.32) | 0.447 |
| FeNO (ppb) | 26 (16.5,38.5) | 25 (14.25,37.75) | 0.553 |
| *Blood routine* | | | |
| WBC (10^9/L) | 6.84 (5.5,8.43) | 7.19 (5.57,9.16) | 0.368 |
| LNR | 4.63 (3.2,6.48) | 5.06 (3,7.05) | 0.499 |
| Eos (10^9/L) | 0.12 (0.06,0.23) | 0.09 (0.03,0.2) | 0.094 |
| Eos% | 2 (0.9,3.75) | 1.2 (0.4,3) | 0.031 |
| *Inflammatory markers* | | | |
| CRP (mg/L) | 5.5 (3.01,14.9) | 12.2 (4.38,30.9) | 0.034 |
| ESR (mm/h) | 20 (5.5,39.75) | 8 (3.05,20) | 0.142 |
| PCT (ng/mL) | 0.1 (0.04,0.16) | 0.1 (0.04,0.13) | 0.801 |
| IL-6 (pg/mL) | 8.52 (3.37,14.77) | 4.24 (2.8,11.67) | 0.343 |
| BNP (ng/L) | 230 (140,1300) | 437 (206,2925) | 0.036 |
| $PaCO_2$ (mmHg) | 45 (41,53) | 46 (44,61) | 0.097 |
| $PaO_2$ (mmHg) | 72 (62,82) | 73 (61,82) | 0.776 |
| *Hepatic and renal function* | | | |
| ALB (g/L) | 36 (33,39) | 34 (31,36) | 0.002 |
| UA (µmol/L) | 303 (223,384) | 303 (248,371) | 0.715 |
| Cr (µmol/L) | 72 (62,98) | 79 (58,93) | 0.965 |
| *Pathogen, n (%)* | 75 (80.65%) | 36 (81.82%) | 0.87 |
| Virus | 13 (20.31%) | 4 (12.12%) | 0.315 |
| Bacteria | 69 (81.18%) | 32 (80%) | 0.876 |
| Fungus | 35 (41.18%) | 22 (61.11%) | 0.045 |
| **FBG** (mmol/L) | 4.8 (3.6, 5.72) | 8.6 (7.25, 10.64) | 0.012 |

[*]Presented as MD (IQR) unless specified otherwise.

FEV1: Forced expiratory volume in 1 s, FVC: forced vital capacity, FeNO: fractional exhaled nitric oxide, WBC: White blood cell, LNR: Lymphocyte to neutrophil ratio, Eos: Eosinophil, CRP: C-reactive protein, ESR: Erythrocyte sedimentation rate, PCT: Procalcitonin, IL-6: interleukin-6, BNP: B-type natriuretic peptide, PCO2: Partial pressure of carbon dioxide, PO2: Partial pressure of oxygen, ALB: Albumin, UA: uric acid, Cr: Creatinine, IQR: interquartile range, MD: median. FBG: fasting blood glucose.

Between the normal blood glucose group and the hyperglycemia group, the proportion of antidiabetic drug use was significantly higher in the hyperglycemia group (p < 0.001). No statistically significant differences were observed in the remaining symptom improvement indicators and treatment outcomes (Table 3). Age (HR = 1.098, 95% CI = 1.013–1.191, p = 0.024), hyperglycemia (HR = 3.622, 95% CI = 1.08–12.15, p = 0.037), and comorbidities with obsolete pulmonary tuberculosis (HR = 3.185, 95% CI = 1.03–9.85, p = 0.044) were identified as independent predictors of mortality in both univariate and multivariate analyses over the follow-up years (Table 4). Hyperglycemia (HR = 1.428, 95% CI = 1.012–2.014, p = 0.043), age (HR = 1.057, 95% CI = 1.04–1.076, p < 0.001), smoking (HR = 1.92, 95% CI = 1.267–2.909, p = 0.002), WBC (HR = 1.063, 95% CI = 1.026–1.101, p = 0.001), Cr (HR = 1.467, 95% CI = 1.23–1.75, p < 0.001), UA (HR = 1.109, 95% CI = 1.017–1.209, p = 0.019), albumin (ALB) (HR = 0.404, 95% CI = 0.255–0.642, p < 0.001), and aspartate aminotransferase (AST) (HR = 1.006, 95% CI = 1.002–1.01, p = 0.004) were also identified as independent predictors of mortality

**Table 3. Comparison of treatment and prognosis between the normal blood glucose group and the hyperglycemia group[*].**

| Items | Normal blood glucose group (N = 101) | Hyperglycemia group (N = 47) | P value |
|---|---|---|---|
| **Characteristics** | | | |
| *Symptom improvement* | | | |
| ΔCAT | 6 (3.75,9.25) | 4 (1,9.75) | 0.377 |
| Δ6MWT(m) | 43 (0,85) | 59 (13,95) | 0.474 |
| ΔmMRC | 2 (1,3) | 2 (1,3) | 0.848 |
| *Treatment and outcomes* | | | |
| Antibiotic therapy, n (%) | 80 (79.21%) | 37 (78.72%) | 0.946 |
| Antibiotic treatment course (days) | 8 (4,11) | 9 (4,10) | 0.907 |
| Hormone therapy, n (%) | 95 (94.06%) | 43 (91.49%) | 0.726 |
| Hormone treatment course (days) | 8 (6,11) | 9 (7,10) | 0.750 |
| Hormone total amount (mg) | 30 (10,50) | 38 (12,52) | 0.434 |
| Antidiabetic medications, n (%) | 7 (6.93%) | 24 (51.06%) | <0.001 |
| Oxygen therapy, n (%) | 83 (82.18%) | 41 (87.23%) | 0.437 |
| Non-invasive positive pressure ventilation, n (%) | 28 (27.72%) | 14 (29.79%) | 0.795 |
| Invasive ventilation, n (%) | 1 (0.99%) | 0 (0%) | 1.000 |
| ICU, n (%) | 6 (5.94%) | 4 (8.51%) | 0.726 |
| Hospitalization days (days) | 9 (8,12) | 10 (8,13) | 0.709 |
| Cost (¥) | 13514 (10312,18358) | 14292 (10635,18341) | 0.697 |
| *Prognosis* | | | |
| *Follow up for one year* | *n = 101* | *n = 47* | |
| Number of exacerbations in the past 12 months | 0 (0,1) | 2 (1.21,2.75) | 0.013 |
| Death, n (%) | 8 (7.92%) | 5 (10.67%) | 0.032 |
| *Follow up for 3 years* | *n = 93* | *n = 42* | |
| Number of exacerbations in the past 12 months | 1 (0.23,1.87) | 3 (2.23,4.21) | 0.007 |
| Death, n (%) | 10 (10.75%) | 9 (21.42%) | 0.019 |
| *Follow up for 5 years* | *n = 83* | *n = 33* | |
| Number of exacerbations in the past 12 months | 3 (2.13,4.27) | 5 (3.90,6.12) | 0.021 |
| Death, n (%) | 14 (16.86%) | 11 (33.33%) | 0.020 |

[*]Presented as MD (IQR) unless specified otherwise.

6MWT: 6-minute walk test, CAT: COPD Assessment Test, mMRC: Modified Medical Research Council, Δ6MWT: 6MWT at discharge – 6MWT at admission, ΔmMRC: mMRC at admission - mMRC at discharge, ΔCAT: CAT at admission – CAT at discharge, ICU: intensive care unit.

**Table 4. COX Regression Analysis of the effect of hyperglycemia on the prognosis of COPD patients with CHD.**

| Variables | Univariate analysis | | Multivariate analysis | |
|---|---|---|---|---|
| | HR (95% CI) | P value | HR (95% CI) | P value |
| Hyperglycemia | 4.426 (1.363,14.376) | 0.013 | 3.622 (1.08,12.15) | 0.037 |
| Age | 1.134 (1.045,1.23) | 0.002 | 1.098 (1.013,1.19) | 0.024 |
| Obsolete pulmonary tuberculosis | 3.816 (1.248,11.666) | 0.019 | 3.185 (1.03,9.852) | 0.044 |
| Sex (male) | 1.357 (1.324,2.543) | 0.021 | 1.33 (0.821,2.658) | 0.120 |
| Smoking | 1.972 (1.47,2.452) | <0.001 | 1.89 (0.987,2.909) | 0.205 |
| Eos | 1.041 (1.013,1.125) | 0.023 | 1.07 (0.841,1.141) | 0.133 |
| ALB | 0.284 (0.198,0.421) | <0.001 | 0.87 (0.791,1.012) | 0.128 |

COPD: Chronic obstructive pulmonary disease, CHD: coronary heart disease, HR: hazard ratio, CI: confidence interval, Eos: Eosinophil, ALB: Albumin.

in univariate and multivariate analyses during the follow-up years, according to NHANES data (Table 5). After a 5-year follow-up period, we compared the mortality rates and incidence of severe acute exacerbations in the previous year between the normal blood glucose group and the hyperglycemia group. The results indicated that during the follow-up period, the incidence of acute exacerbations and mortality rates were higher in the hyperglycemia group compared to the normal blood glucose group (Table 3). Kaplan–Meier survival curves demonstrated that patients in the hyperglycemia group experienced significantly higher rates of exacerbations and mortality compared to those in the normal blood glucose group over a follow-up period of 5 years or more (log-rank test, p = 0.013, p = 0.024, and p = 0.016 respectively; Fig 2). The Kaplan-Meier survival curves demonstrated that during our 12-month follow-up period, patients in the hyperglycemia group exhibited significantly higher rates of acute disease exacerbation compared to those in the normal blood glucose group (log-rank test, p = 0.013; Fig 2A). Over the extended follow-up period exceeding 60 months, the hyperglycemia group showed markedly increased mortality relative to the normal blood glucose group (log-rank test, p = 0.024; Fig 2B). Furthermore, our analysis of the NHANES database revealed that during an extended observation period of over 200 months, mortality rates remained significantly elevated in the hyperglycemic cohort compared to the normoglycemic controls (log-rank test, p = 0.016; Fig 2C).

## Discussion

This study preliminarily investigates the impact of hyperglycemia on the clinical prognosis of patients with COPD and CHD, leveraging a robust combination of local datasets and the extensive NHANES database. The findings reveal a significantly reduced long-term survival rate post-discharge in the hyperglycemia cohort among patients with COPD and CHD. Although the detrimental effects of hyperglycemia on clinical outcomes in patients with either COPD or CHD alone are well-established, its prognostic impact in the context of COPD-CHD comorbidity remains poorly characterized. Emerging evidence suggests that hyperglycemia may potentiate the pathophysiological interplay between these conditions, potentially through amplification of shared mechanisms including chronic low-grade inflammation, endothelial dysfunction, and metabolic dysregulation [17–19]. To date, no research has comprehensively examined the relationship between hyperglycemia and the dual conditions of COPD and CHD. This study is pioneering in identifying hyperglycemia as a potential independent risk factor for long-term mortality in patients with concurrent COPD and CHD.

The long-term adverse prognostic implications of hyperglycemia in patients concurrently suffering from both COPD and CHD are likely attributable to the intricate interplay between hyperglycemia, CHD, and COPD. CHD is a prevalent

**Table 5. COX Regression Analysis of the effect of hyperglycemia on the prognosis of COPD patients with CHD from NHANES.**

| Variables | Univariate analysis | | Multivariate analysis | |
|---|---|---|---|---|
| | HR (95% CI) | P value | HR (95% CI) | P value |
| Hyperglycemia | 1.505 (1.08,2.096) | 0.016 | 1.428 (1.012,2.014) | 0.043 |
| Sex (male) | 1.657 (1.223,2.243) | 0.001 | 1.275 (0.924,1.758) | 0.139 |
| Age | 1.062 (1.046,1.078) | <0.001 | 1.057 (1.04,1.076) | <0.001 |
| Smoking | 2.177 (1.47,3.222) | <0.001 | 1.92 (1.267,2.909) | 0.002 |
| Eos | 1.055 (1.012,1.1) | 0.012 | 1.063 (1.026,1.101) | 0.001 |
| Cr | 1.662 (1.475,1.873) | <0.001 | 1.467 (1.23,1.75) | <0.001 |
| UA | 1.201 (1.108,1.302) | <0.001 | 1.109 (1.017,1.209) | 0.019 |
| ALB | 0.314 (0.208,0.474) | <0.001 | 0.404 (0.255,0.642) | <0.001 |
| AST | 1.005 (1.001,1.009) | 0.025 | 1.006 (1.002,1.01) | 0.004 |

COPD: Chronic obstructive pulmonary disease, CHD: coronary heart disease, NHANES: National Health and Nutrition Examination Survey, HR: hazard ratio, CI: confidence interval, Eos: Eosinophil, ALB: Albumin, UA: uric acid, Cr: Creatinine, AST: glutamic oxaloacetic transaminas.

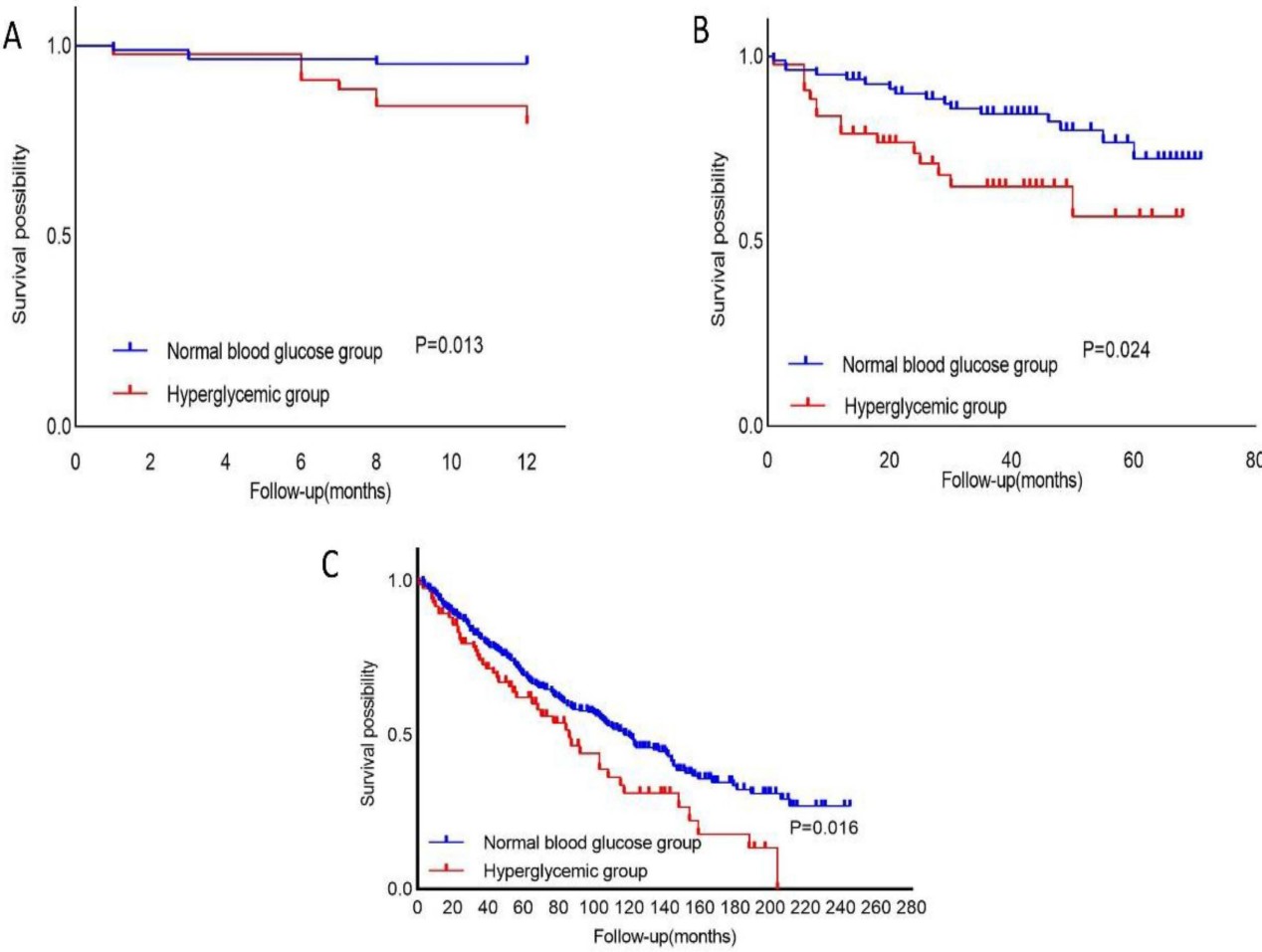

**Fig 2. Kaplan-Meier curves for time to moderate-to-severe exacerbation (A), all-cause mortality (B), and NHANES-derived mortality (C) in patients with hyperglycemia and normal blood glucose groups during follow-up.** NHANES: National Health and Nutrition Examination Survey.

comorbidity among COPD patients, sharing numerous risk factors, including smoking, aging, dietary habits, and environmental pollutants such as air pollution [17]. Both CHD and COPD are intricately linked to inflammatory processes, with hyperglycemia potentially exacerbating these by promoting the activation of nuclear factor-kB (NF-κB), thus triggering airway inflammation [18]. Additionally, hyperglycemia may precipitate mitochondrial dysfunction, thereby enhancing the production of mitochondrial free radicals, reactive oxygen species, and pro-inflammatory cytokines, culminating in elevated systemic inflammatory levels [19,20]. Chronic hyperglycemia markedly exacerbates systemic oxidative stress. Exposure to smoke or other particulate pollutants can result in substantial damage to pulmonary cells, augment mucus secretion in airway epithelial cells, and stimulate the production of significant quantities of inflammatory mediators, including tumor necrosis factor-α (TNF-α), IL-1β, and IL-6. Concurrently, excessive mucus secretion and neutrophil aggregation activate a multitude of inflammatory mediators, thereby generating increased levels of ROS and further exacerbating oxidative stress [21–23]. Oxidative stress amplifies the body's inflammatory response by modulating NF-κB and activator protein-1 (AP-1), leading to the release of substantial quantities of cytokines, including IL-1β and TNF-α [24]. When hyperglycemia, COPD, and CHD concurrently afflict the body, elevated blood glucose levels may further intensify the systemic inflammatory response, thereby worsening the clinical course of COPD.

Hyperglycemia exerts a direct deleterious impact on the cardiovascular system. Prolonged hyperglycemia precipitates vascular endothelial dysfunction, atherosclerosis, and a spectrum of other pathological alterations. The direct endothelial damage induced by hyperglycemia can potentiate pulmonary vascular disease and precipitate a decline in pulmonary function [25]. The deterioration of lung function and ensuing hypoxia in COPD patients can exacerbate the cardiovascular damage due to hyperglycemia, with the worsening of cardiovascular disease serving as a catalyst for acute COPD exacerbations [17]. Moreover, certain therapeutic agents for CHD, notably β-blockers, may induce bronchoconstriction, triggering moderate to severe asthma attacks, thereby heightening the mortality risk in COPD patients. Early administration of β-blockers can precipitate a subtle decline in pulmonary function [26]. In summary, the coexistence of hyperglycemia, COPD, and CHD may potentiate the interplay between COPD and CHD.

Glucagon-Like Peptide 1 (GLP-1) receptor agonists and Dipeptidyl Peptidase 4 (DPP-4) inhibitors, commonly employed in diabetes management, have been shown to reduce cardiovascular risk in patients, thereby potentially mitigating the deleterious effects of COPD [27,28]. Moreover, GLP-1 receptor agonists have been demonstrated to promote bronchial relaxation and enhance airway function. In addition, certain antidiabetic agents exhibit anti-inflammatory properties, which may contribute to reducing pulmonary inflammation and ameliorating the clinical status of COPD [26].

COPD forms a unique pathophysiological network through the systemic effects of amplified systemic inflammation, hypoxia-oxidative stress crosstalk, and pulmonary vascular remodeling, thereby significantly potentiating the pathogenic role of hyperglycemia in CHD. Patients with COPD exhibit a persistent state of systemic inflammation, characterized by markedly elevated levels of proinflammatory cytokines such as IL-6 and TNF-α in the circulation. This chronic inflammation reinforces the pathogenic impact of hyperglycemia on CHD through two distinct pathways: metabolic derangement and accelerated vascular damage [29]. Studies have demonstrated that coronary artery plaques in COPD patients are more prone to calcification and instability, with hyperglycemia further increasing the risk of plaque rupture [30]. Furthermore, the characteristic pathological changes of COPD (e.g., emphysema, small airway remodeling) lead to chronic hypoxia, which, in conjunction with hyperglycemia, constitutes a dual hit to oxidative stress. Activation of hypoxia-inducible factors (HIFs) accelerates the proliferation of coronary artery smooth muscle cells and intimal thickening [31]. Additionally, research has revealed that pulmonary vascular remodeling results in pulmonary hypertension, compelling the right ventricle to increase its output. Such hemodynamic abnormalities severely impair coronary perfusion and augment cardiac workload [32].

Our research demonstrates that hyperglycemia is linked to an adverse long-term prognosis in patients suffering from both COPD and CHD, with these findings bearing substantial clinical implications. Nevertheless, this study is not without limitations. First and foremost, we incorporated NHANES data into our analysis. However, the relatively small sample size of our single-center Chinese cohort limits the generalizability of these specific findings. Furthermore, given the inherent differences in the variables initially included in the design of the two databases, there are discrepancies in the prognostic risk factors identified between the two datasets. Nevertheless, both databases consistently confirm that hyperglycemia is an independent risk factor for acute exacerbations and mortality. In subsequent follow-up studies, we will continue to expand our database to minimize such discrepancies. Secondly, the study does not sufficiently consider the potential confounding effects of diabetes or hypoglycemic agents on the observed outcomes. Moreover, the absence of data regarding the duration of both COPD and CHD could potentially introduce bias into the study's findings. Future research endeavors will be directed towards exploring the role of hypoglycemic agents and evaluating the impact of glycemic control on the prognosis of patients with COPD and CHD. Third, incorporating cohorts of patients with COPD alone or CHD alone into the analysis would enhance the comprehensiveness of our findings and better validate whether the observed prognostic effects arise from the synergistic interaction induced by the coexistence of the two diseases. Moving forward, we will design prospective studies to systematically compare the clinical characteristics and prognostic disparities between patients with comorbid COPD and CHD versus those with either disease alone.

## Conclusion

Hyperglycemia serves as an independent risk factor for prolonged acute exacerbations and mortality in COPD patients complicated with CHD post-discharge. Proactive intervention strategies targeting hyperglycemia should be promptly instituted to mitigate the risk of future acute exacerbations and mortality in COPD patients with CHD.

## Author contributions

**Conceptualization:** Yan Chen.

**Data curation:** Zhongshang Dai, Chenjie He, Huiui Zeng.

**Formal analysis:** Huiui Zeng.

**Funding acquisition:** Zhongshang Dai, Yan Chen.

**Software:** Huiui Zeng.

**Writing – original draft:** Zhongshang Dai, Chenjie He.

**Writing – review & editing:** Yan Chen.

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
