## [Decision Letter · Decision Letter 0]

2 Jul 2025

Dear Dr. Yen,

Thank you for submitting your manuscript to PLOS ONE. After careful consideration, we feel that it has merit but does not fully meet PLOS ONE’s publication criteria as it currently stands. Therefore, we invite you to submit a revised version of the manuscript that addresses the points raised during the review process.

**ACADEMIC EDITOR: **

Thank you for submitting your valuable work to PLOS ONE. One of the reviewers has requested a clear rationale for focusing on patients with both COPD and CHD, as well as a comparative analysis of each condition. As the editor, I agree with this comment and look forward to your response. I am looking forward to receiving your revised manuscript

We look forward to receiving your revised manuscript.

Kind regards,

Hidetaka Hamasaki

Academic Editor

PLOS ONE

Additional Editor Comments (if provided):

Reviewers' comments:

Reviewer's Responses to Questions

**Comments to the Author**

1. Is the manuscript technically sound, and do the data support the conclusions?

Reviewer #1: No

Reviewer #2: No

Reviewer #3: No

2. Has the statistical analysis been performed appropriately and rigorously?

Reviewer #1: No

Reviewer #2: Yes

Reviewer #3: No

3. Have the authors made all data underlying the findings in their manuscript fully available?

Reviewer #1: Yes

Reviewer #2: Yes

Reviewer #3: Yes

4. Is the manuscript presented in an intelligible fashion and written in standard English?

Reviewer #1: Yes

Reviewer #2: Yes

Reviewer #3: Yes

Reviewer #1: Dear Authors,

I appreciate the opportunity to review your submitted manuscript. Your work addresses an interesting and clinically informative topic regarding the prognosis of patients with co-existing COPD and CHD in the context of hyperglycemia. However, I have identified several significant points that require substantial addressing to enhance the scientific rigor and clinical interpretability of your findings.

1. The rationale for specifically examining patients with both COPD and CHD, as opposed to those with either condition alone, needs to be more clearly articulated. While the individual impact of hyperglycemia on the prognosis of COPD or CHD is well-documented, the unique clinical implications of hyperglycemia in the presence of both conditions require further explanation to underscore the study's specific contribution.

2. A critical limitation noted is the absence of a comparative analysis with cohorts of patients having either COPD alone or CHD alone. Without such a comparison, it remains challenging to definitively ascertain whether the observed prognostic effects are genuinely synergistic due to the co-existence of both diseases or simply reflect the known impact of hyperglycemia on one of the conditions. The discussion, while comprehensive regarding COPD and hyperglycemia, might benefit from a deeper exploration of how the interrelationship between hyperglycemia and CHD is uniquely modulated by the presence of COPD, beyond the already established associations. This comparative perspective is essential to conclude that the combined disease state is the true driver of your reported outcomes.

3. While the inclusion of NHANES data is commendable, the relatively small sample size from the single-center Chinese cohort limits the generalizability of these specific findings. The differing prognostic factors identified between your local data and NHANES suggest potential heterogeneity that warrants careful consideration in your interpretation.

Reviewer #2: The authors have investigated the impact of hyperglicemia on the prognosis of patients with COPD and CHD in a 5 year study analysis. The paper is well presented, however I have some concern regarding how well were the patients with diabetes investigated. In table 1 are given the general characteristics of patients compared by the level of fasting glycemia at admission. In the normal glycemic group, it was stated that 13.86% of them had diabetes, in the other group were patients with > 7mmol/l glycemia, a value that diagnoses diabetes, but only 36.17% were diagnosed previously with diabetes. Please explain this disconcordance. This is a major concern. There is no data about HbA1c or diabetes duration, diabetes treatment. If the whole patients had diabetes, then a comparison regarding long-term glycemic control reflected by HbA1c would have been indicated. Otherwise, a comparison between patients with or without diabetes is suggested. Also, figure 2 is not fully supported by statistical data, please check again the interpretation.

Reviewer #3: Dear Authors,

This research addresses a crucial subject: the impact of hyperglycemia on outcomes and prognosis in COPD and CHD. However, the methods described require clarification to provide a more straightforward conclusion.

#1 It is unclear whether the authors intended to evaluate the influence of stress-induced hyperglycemia or chronic exposure to hyperglycemia ( decompensated diabetes). If it is stress-induced hyperglycemia, some studies have reported that it is related to worse outcomes in the ICU, and the stress hyperglycemia ratio is better at distinguishing it from diabetes hyperglycemia. Furthermore, it is challenging to diagnose stress-hyperglycemia with only an admission fasting glucose.

#2 The definitions of the two groups for comparison included only an admission fasting glucose level of 7 mmol/L, and there is no clear explanation of how the patients were classified as having diabetes or not (in both groups, less than 20% reported having diabetes). There is no data on HbA1C, the use of drugs to treat or other methods to identify diabetes.

#3 Despite the description of methods not including the collection of data on glycemia or HbA1C during the follow-up, the discussion is based on the influence of chronic exposure to hyperglycemia. Moreover, chronic hyperglycemia is decompensated diabetes. The presence of diabetes was reinforced by the authors when they suggested the use of drugs for the treatment of diabetes to control hyperglycemia. Thus, it is necessary to review the percentage of patients with diabetes and/or add some comments about that.

**Do you want your identity to be public for this peer review?** For information about this choice, including consent withdrawal, please see our Privacy Policy

Reviewer #1: **Yes: ** KYU YONG CHO

Reviewer #2: No

Reviewer #3: No

---

## [Author Response · Author response to Decision Letter 1]

31 Jul 2025

Reviewer1

We sincerely appreciate the opportunity to revise our manuscript and are deeply grateful for the reviewers' insightful comments, which have significantly helped improve the quality of our work. We have revised the article according to the comments of the reviewers and editors, and marked the modified parts in red font. The following are specific changes based on your suggestions.

SPECIFIC COMMENTS:

1. The rationale for specifically examining patients with both COPD and CHD, as opposed to those with either condition alone, needs to be more clearly articulated.

We sincerely appreciate this invaluable comment. We apologize for the insufficient clarity in articulating the rationale for focusing on patients with both COPD and CHD. To address this, we have expanded the introduction and discussion sections to emphasize the unique clinical significance of this comorbid population. Epidemiological and pathophysiological rationale: COPD and CHD share common pathophysiological mechanisms and risk factors, including smoking, aging, and chronic systemic inflammation, leading to a high prevalence of comorbidity in elderly populations (20–30%). This coexistence is not merely additive but synergistic: acute COPD exacerbations may precipitate myocardial ischemia through hypoxemia, sympathetic activation, and increased ventricular afterload, whereas impaired cardiac output in CHD can exacerbate ventilatory-perfusion mismatch, worsening COPD-associated hypoxia and hypercapnia [Introduction, Lines 14-22]. Moreover, as you rightly pointed out, while the impact of hyperglycemia on the prognosis of patients with either COPD or CHD alone has been well-documented, its unique clinical implications in the context of comorbid COPD and CHD remain insufficiently explained. We greatly appreciate this valuable insight and have now incorporated a detailed analysis of this issue in the Discussion section [Lines 6-13] of our manuscript. Although the detrimental effects of hyperglycemia on clinical outcomes in patients with either COPD or CHD alone are well-established, its prognostic impact in the context of COPD-CHD comorbidity remains poorly characterized. Emerging evidence suggests that hyperglycemia may potentiate the pathophysiological interplay between these conditions, potentially through amplification of shared mechanisms including chronic low-grade inflammation, endothelial dysfunction, and metabolic dysregulation.

2. A limitation noted is the absence of a comparative analysis with cohorts of patients having either COPD alone or CHD alone.

We sincerely appreciate your insightful question, which was indeed a critical aspect we carefully considered during the initial design of our study. To begin with, our team collectively confirmed that the core focus of our research is to investigate the impact of hyperglycemia on the prognosis of patients with concurrent COPD and CHD. Through our preliminary literature review, we found that numerous studies have explored the effects of hyperglycemia in patients with either COPD alone or CHD alone, consistently concluding that hyperglycemia exerts significant adverse effects on disease progression and prognosis in these single-disease populations. Based on this well-established evidence, we acknowledged and defaulted to these prior findings, thus not including cohorts of patients with isolated COPD or isolated CHD to analyze the impact of blood glucose in the current study. However, research on the influence of hyperglycemia and its prognostic implications in patients with concurrent COPD and CHD remains largely unexplored and unreported in the existing literature. This identified gap underscores the innovative value of our chosen topic. We subsequently stratified patients with comorbid COPD and CHD into hyperglycemic and normoglycemic groups, and analyzed differences in baseline demographics, laboratory parameters, clinical characteristics, one-year acute exacerbation rates, and mortality between the two groups. This approach enabled us to derive clinically meaningful findings regarding the prognostic role of hyperglycemia in this comorbid population. Nevertheless, we fully agree with your valuable suggestion that incorporating cohorts of patients with isolated COPD or CHD would significantly enhance the comprehensiveness of our results, thereby better validating whether the observed prognostic effects arise from the synergistic interaction of the two coexisting diseases. We sincerely thank you for this constructive reminder. We have explicitly addressed this limitation in the discussion section of the manuscript [Discussion, The last paragraph, lines 12-19]. Furthermore, we aim to build upon your insights by designing prospective studies in the future to systematically compare the clinical characteristics and prognostic differences between patients with comorbid COPD+CHD and those with either disease alone.

Furthermore, you also suggested the need for a more in-depth exploration of how the presence of COPD uniquely modulates the interplay between hyperglycemia and CHD. We greatly appreciate this insightful recommendation. In response, we have conducted a renewed search of the latest literature in accordance with your feedback, delved deeper into the mechanisms underlying the unique modulation of the hyperglycemia-CHD relationship by COPD, and incorporated these findings into the discussion section of our manuscript [Discussion, Penultimate paragraph]. COPD forms a unique pathophysiological network through the systemic effects of amplified systemic inflammation, hypoxia-oxidative stress crosstalk, and pulmonary vascular remodeling, thereby significantly potentiating the pathogenic role of hyperglycemia in CHD. Patients with COPD exhibit a persistent state of systemic inflammation, characterized by markedly elevated levels of proinflammatory cytokines such as IL-6 and TNF-α in the circulation. This chronic inflammation reinforces the pathogenic impact of hyperglycemia on CHD through two distinct pathways: metabolic derangement and accelerated vascular damage29. Studies have demonstrated that coronary artery plaques in COPD patients are more prone to calcification and instability, with hyperglycemia further increasing the risk of plaque rupture30. Furthermore, the characteristic pathological changes of COPD (e.g., emphysema, small airway remodeling) lead to chronic hypoxia, which, in conjunction with hyperglycemia, constitutes a dual hit to oxidative stress. Activation of hypoxia-inducible factors (HIFs) accelerates the proliferation of coronary artery smooth muscle cells and intimal thickening31. Additionally, research has revealed that pulmonary vascular remodeling results in pulmonary hypertension, compelling the right ventricle to increase its output. Such hemodynamic abnormalities severely impair coronary perfusion and augment cardiac workload32.

3. While the inclusion of NHANES data is commendable, the relatively small sample size from the single-center Chinese cohort limits the generalizability of these specific findings.

We sincerely appreciate this constructive reminder. The inclusion of NHANES data in our analysis was intended to compare with our local database, aiming to examine the prognostic impact of hyperglycemia in patients with concurrent COPD and CHD. Fortunately, our findings were consistent across both datasets: during the follow-up period, the incidence of acute exacerbations and mortality were significantly higher in the hyperglycemic group than in the normoglycemic group, with hyperglycemia identified as an independent risk factor for both outcomes in both our local cohort and the NHANES dataset. However, as aptly noted, the relatively small sample size of our single-center Chinese cohort limits the generalizability of these specific findings, which represents a key limitation of our study. Additionally, inherent differences in the baseline variables collected during the initial design of the two databases contributed to discrepancies in the prognostic factors identified between the analyses. Nevertheless, both datasets convergently confirmed that hyperglycemia is an independent risk factor for acute exacerbations and mortality. We greatly value your identification of these limitations, and we have incorporated these points into the discussion section of the manuscript as per your suggestions [Discussion, The sixth paragraph, lines 5-13]. In subsequent follow-up studies, we will continue to expand our database to minimize such discrepancies.

All in all, we really appreciate your being so serious in reviewing the manuscript and giving so many precious suggestions. We have benefited a lot. We have made a lot of efforts to get this study published, and we hope to get your support. Thank you so much.

Reviewer 2

We sincerely appreciate the opportunity to revise our manuscript and are deeply grateful for the reviewers' insightful comments, which have significantly helped improve the quality of our work. We have revised the article according to the comments of the reviewers and editors, and marked the modified parts in red font. The following are specific changes based on your suggestions.

SPECIFIC COMMENTS:

1. In the normal glycemic group, it was stated that 13.86% of them had diabetes, in the other group were patients with > 7mmol/l glycemia, a value that diagnoses diabetes, but only 36.17% were diagnosed previously with diabetes. Please explain this disconcordance.

We are particularly grateful for your identification of this valuable and meaningful issue. First and foremost, we assure you that all our clinical data are derived from patients' real - world information and medical records, ensuring their authenticity and reliability. However, when we analyzed this result, we were also filled with confusion and even hesitated whether to include it in the table, worrying that it might cause ambiguity. After analyzing this result as a team, we have the following explanations. In our study cohort, patients were divided into the normoglycemic group and the hyperglycemic group based on whether their FBG exceeded 7 mmol/L. The presence of diabetes in the normoglycemic group may be due to the fact that these patients had good blood glucose control, so their measured FBG did not exceed 7 mmol/L. Regarding the phenomenon that not all patients in the hyperglycemic group are diabetic, it should be considered that some patients may have stress - induced hyperglycemia, and hyperglycemia does not mean that all of them are diabetic patients. We sincerely appreciate your meticulousness and kindness in detecting and reminding us of this confusing issue. To avoid further ambiguity, we consider it more appropriate to delete the row of "Diabetes" in Table 1. We hope that our explanation and correction will meet your approval and satisfaction.

2. There is no data about HbA1c or diabetes duration, diabetes treatment. Figure 2 is not fully supported by statistical data, please check again the interpretation.

We greatly appreciate your excellent question. This limitation was briefly addressed in the discussion section of our manuscript, and we would like to elaborate further here. In our hospital, blood glucose levels are routinely measured via fasting capillary blood glucose testing. Typically, patients do not undergo a second measurement unless their fasting blood glucose exceeds 7 mmol/L, in which case a repeat test or HbA1c measurement is performed. Due to regulations set by the National Healthcare Security Administration, HbA1c is not a routine test and is only measured in patients with hyperglycemia or a confirmed diagnosis of diabetes. Regrettably, this prevented us from collecting HbA1c data for all patients to compare long-term glycemic control. However, in subsequent follow-up studies, we will expand our database and prospectively collect data on HbA1c, diabetes duration, and diabetes treatment regimens for further analysis. Additionally, you put forward an excellent suggestion to compare patient cohorts with and without diabetes. We are pleased to inform you that this analysis has been developed into a separate manuscript for submission, in which we found that COPD patients with comorbid diabetes have poorer prognoses. We hope that this subsequent study will have the opportunity to benefit from your guidance again and be successfully published. Furthermore, you kindly pointed out that the description of Figure 2 is insufficiently rigorous. We appreciate your meticulous reminder, as this indeed reflects an oversight on our part. We have revised the results section of the manuscript and re-described the content of Figure 2 accordingly. The Kaplan-Meier survival curves demonstrated that during our 12-month follow-up period, patients in the hyperglycemia group exhibited significantly higher rates of acute disease exacerbation compared to those in the normal blood glucose group (log-rank test, p=0.013; Figure 2A). Over the extended follow-up period exceeding 60 months, the hyperglycemia group showed markedly increased mortality relative to the normal blood glucose group (log-rank test, p=0.024; Figure 2B). Furthermore, our analysis of the NHANES database revealed that during an extended observation period of over 200 months, mortality rates remained significantly elevated in the hyperglycemic cohort compared to the normoglycemic controls (log-rank test, p=0.016; Figure 2C) [Results, The last paragraph, lines 28-39]. Please accept our sincere thanks once again.

All in all, we really appreciate your being so serious in reviewing the manuscript and giving so many precious suggestions. We have benefited a lot. We have made a lot of efforts to get this study published, and we hope to get your support. Thank you so much.

Reviewer 3

We sincerely appreciate the opportunity to revise our manuscript and are deeply grateful for the reviewers' insightful comments, which have significantly helped improve the quality of our work. We have revised the article according to the comments of the reviewers and editors, and marked the modified parts in red font. The following are specific changes based on your suggestions.

SPECIFIC COMMENTS:

1. It is unclear whether the authors intended to evaluate the influence of stress-induced hyperglycemia or chronic exposure to hyperglycemia ( decompensated diabetes).

Thank you sincerely for your insightful comments, which have helped us clarify critical methodological and interpretive issues in our manuscript. We appreciate your clarification on distinguishing stress-induced hyperglycemia from chronic hyperglycemia (decompensated diabetes). Our initial objective was to evaluate the prognostic impact of admission hyperglycemia (encompassing both stress-induced and chronic hyperglycemia) on patients with COPD and CHD, given that both phenotypes are common in acute hospital settings and linked to poor outcomes in comorbid populations. We acknowledge the limitation of using only admission fasting glucose to differentiate these subtypes. In our cohort, stress-induced hyperglycemia (e.g., due to acute exacerbations of COPD or cardiac events) likely contributed to the hyperglycemia observed in non-diabetic patients, while chronic hyperglycemia reflected underlying diabetes. However, due to the retrospective nature of our data, we lacked systematic measurements of stress hyperglycemia ratios (e.g., glucose/insulin ratio) or pre-admission glycemic profiles to rigorously distinguish these entities—this is a key limitation we have now explicitly noted in the discussion. Moving forward, we agree that incorporating stress hyperglycemia ratios and pre-admission HbA1c would enhance subtype classification, and we plan to include these metrics in future prospective studies.

2. The definitions of the two groups for comparison included only an admission fasting glucose level of 7 mmol/L, and there is no clear explanation of how the patients were classified as having diabetes or not.

We greatly appreciate your excellent question. This limitation was briefly addressed in the discussion section of our manuscript, and we would like to elaborate further here. In our hospital, blood glucose levels are routinely measured via fasting capillary blood glucose testing. Typically, patients do not undergo a second measureme

---

## [Decision Letter · Decision Letter 1]

24 Aug 2025

Dear Dr. Yen,

Thank you for submitting your manuscript to PLOS ONE. After careful consideration, we feel that it has merit but does not fully meet PLOS ONE’s publication criteria as it currently stands. Therefore, we invite you to submit a revised version of the manuscript that addresses the points raised during the review process.

We look forward to receiving your revised manuscript.

Kind regards,

Hidetaka Hamasaki

Academic Editor

PLOS ONE

Journal Requirements:

Reviewers' comments:

Reviewer's Responses to Questions

**Comments to the Author**

Reviewer #1: All comments have been addressed

Reviewer #3: All comments have been addressed

2. Is the manuscript technically sound, and do the data support the conclusions?

Reviewer #1: Yes

Reviewer #3: Partly

3. Has the statistical analysis been performed appropriately and rigorously?

Reviewer #1: Yes

Reviewer #3: No

4. Have the authors made all data underlying the findings in their manuscript fully available?

Reviewer #1: Yes

Reviewer #3: Yes

5. Is the manuscript presented in an intelligible fashion and written in standard English?

Reviewer #1: Yes

Reviewer #3: Yes

Reviewer #1: Dear Authors,

Thank you for your detailed and thoughtful response. Your answer completely clarifies my question.

I have no further questions and comments.

Reviewer #3: Dear Authors,

I am glad to rereview this manuscript. The authors addressed the reviewers' comments, but some still require revision.

#1 Clarify the Methods - The abstract states that recruitment occurred in December 2016, with follow-up until March 2023. However, the Methods item Study population indicates that participant recruitment spanned from December 30, 2016, to March 5, 2023.

#2 Also in the Methods, item Study design says: "All patients were followed up for a period exceeding 12 months." Moreover, in Table 3, there are some data for exacerbations and mortality for 1, 3, and 5 years. It would be beneficial to detail the methodology and to include data on the number of participants who were followed during such periods.

It would be valuable to detail in the methods how often trained research assistants collected the data by telephone consultations with patients and their relatives, interviews in outpatient clinics, or reviews of medical records provided by patients.

#3 Table 3 presents the number of exacerbations and deaths in 1, 3, and 5 years. However, it lacks Kaplan-Meier curves for the study sample, as the subtitle suggests that Figure 2 pertains to the NHANES sample.

#4 In the discussion, the authors include some comments on the influence of some antidiabetic medications; however, despite the methods stating that the researchers collected the data using 'Self-administered questionnaires, clinical records, and self-reported data there seems to be no information on diabetes treatment or glicemic control that would be very important for discussion could be obtained.

#5 Avoid using definitive assertions such as "This study rigorously investigates the impact of hyperglycemia" and "Our research unequivocally demonstrates" as the methods do not allow such affirmation, considering the diagnosis of hyperglycemia in the hospital of the study uses routinely for glucose level evaluation a fasting capillary blood test. Add comments on whether there were data on results confirmation with blood samples. Also, it is important to include data for the glucose levels in the results tables, such as the mean levels.

**Do you want your identity to be public for this peer review?** For information about this choice, including consent withdrawal, please see our Privacy Policy

Reviewer #1: No

Reviewer #3: No

---

## [Author Response · Author response to Decision Letter 2]

8 Oct 2025

Reviewer #3

We sincerely appreciate the opportunity to revise our manuscript and are deeply grateful for the reviewers' insightful comments, which have significantly helped improve the quality of our work. We have revised the article according to the comments of the reviewers and editors, and marked the modified parts in red font. The following are specific changes based on your suggestions.

SPECIFIC COMMENTS:

1. Clarify the Methods - The abstract states that recruitment occurred in December 2016, with follow-up until March 2023. However, the Methods item Study population indicates that participant recruitment spanned from December 30, 2016, to March 5, 2023.

Thank you very much for your kind reminder; this was an oversight on our part. As per your suggestion, we have revised the sentence in the Methods section from " The participant recruitment period spanned from 30/12/2016 to 05/03/2023" to align with the Abstract: " Recruitment for the study commenced in December 30, 2016, with follow-up until March 5, 2023. " Once again, we sincerely appreciate your careful attention in pointing out this error.

2. All patients were followed up for a period exceeding 12 months. It would be beneficial to detail the methodology and to include data on the number of participants who were followed during such periods.

Thank you very much for raising this question regarding the follow-up process; it has been truly helpful to us. We acknowledge that our Methods section did not provide sufficient detail on the implementation of follow-up at each time point, and the table also omitted the specific numbers of participants followed up. We have now supplemented both sections based on your kind reminder and suggestions. Specifically, we have added a detailed description of the follow-up procedures at each time point in the Methods section“We collected data on acute exacerbations of AECOPD during the first, third, and fifth years of follow-up, as well as all-cause mortality data throughout the entire follow-up period. The specific follow-up process was conducted by trained research assistants through telephone consultations or outpatient interviews with patients and their relatives (at least once per year). Data on annual exacerbations and survival status, along with outpatient and inpatient medical records provided by the patients, were uniformly recorded in a database, with data collected every three months.”. Additionally, we have included the exact number of participants followed up at each time point in Table 3. We greatly appreciate your help in making our research methodology more rigorous and complete.

3. Table 3 presents the number of exacerbations and deaths in 1, 3, and 5 years. However, it lacks Kaplan-Meier curves for the study sample, as the subtitle suggests that Figure 2 pertains to the NHANES sample.

Thank you very much for your kind reminder and suggestions. We suspect this is a beautiful misunderstanding. We would like to explain to you that Figure 2 is actually based on the results of both our study and the analysis of the NHANES database. Figures 2A and 2B are the Kaplan-Meier curves for the study sample, and Figure 2C is the Kaplan-Meier curve based on the NHANES sample. However, it might be that the previous title of Figure 2 was not clear and detailed enough, which led to your misunderstanding. We have now revised the title of Figure 2 to: "Figure 2. Kaplan-Meier curves for time to moderate-to-severe exacerbation (A), all-cause mortality (B), and NHANES-derived mortality (C) in patients with hyperglycemia and normal blood glucose groups during follow-up." Thank you again for pointing out this issue.

4. In the discussion, the authors include some comments on the influence of some antidiabetic medications; however, despite the methods stating that the researchers collected the data using 'Self-administered questionnaires, clinical records, and self-reported data there seems to be no information on diabetes treatment that would be very important for discussion could be obtained.

Thank you for your excellent suggestion, which very aptly pointed out the shortcomings in our work. As you rightly noted, the discussion section of our study does mention the impact of antidiabetic drugs, making data on diabetes treatment regimens crucial to that part of the discussion. Following your advice, we have supplemented Table 3 with information on the use of antidiabetic medications by patients in both the normal blood glucose group and the hyperglycemia group to make the research data more comprehensive. Additionally, we have correspondingly revised the description in the 'Results' section of the article [Results section, fourth paragraph, lines 1-5]. “Between the normal blood glucose group and the hyperglycemia group, the proportion of antidiabetic drug use was significantly higher in the hyperglycemia group (p < 0.001). No statistically significant differences were observed in the remaining symptom improvement indicators and treatment outcomes (Table 3).” We hope that our revisions now meet with your approval. Thank you once again!

5. Avoid using definitive assertions such as "This study rigorously investigates the impact of hyperglycemia" and "Our research unequivocally demonstrates" as the methods do not allow such affirmation, considering the diagnosis of hyperglycemia in the hospital of the study uses routinely for glucose level evaluation a fasting capillary blood test.

We sincerely thank you for providing such thoughtful and genuine feedback. The absolute statements used in our discussion were indeed inaccurate and inappropriate. We greatly appreciate you pointing out this shortcoming in a timely manner. We have revised the text according to your suggestions: “This study rigorously investigates the impact of hyperglycemia” has been changed to “This study preliminarily investigates the impact of hyperglycemia,” and “Our research unequivocally demonstrates” has been revised to “Our research demonstrates”. You also mentioned whether there is data to verify the diagnosis of hyperglycemia through blood tests. This limitation was briefly addressed in the discussion section of our manuscript, and we would like to elaborate further here. In our hospital, blood glucose levels are routinely measured via fasting capillary blood glucose testing. Typically, patients do not undergo a second measurement unless their fasting blood glucose exceeds 7 mmol/L, in which case a repeat test or HbA1c measurement is performed. Due to regulations set by the National Healthcare Security Administration, HbA1c is not a routine test and is only measured in patients with hyperglycemia or a confirmed diagnosis of diabetes. Regrettably, this prevented us from collecting HbA1c data for all patients to compare long-term glycemic control. However, in subsequent follow-up studies, we will expand our database and prospectively collect data on HbA1c for further analysis. Finally, you also suggested adding data related to blood glucose levels in the table of the results section. This was indeed an oversight on our part, and we have now included the median fasting blood glucose data for both the normoglycemic and hyperglycemic patient groups in Table 2. We have also incorporated these findings into the main text [Results section, Paragraph 3, Lines 11–13]: “Additionally, the FBG levels in the hyperglycemia group were significantly higher than those in the normal blood glucose group (p = 0.012)”. We really appreciate your being so serious in reviewing the manuscript and giving so many precious suggestions. We have benefited a lot.

We are truly touched that, even during this second round of review, you have provided us with such valuable suggestions to help us continuously improve the quality of our paper. We are deeply grateful to you and sincerely hope to earn your approval this time. Thank you so much.

---

## [Decision Letter · Decision Letter 2]

20 Oct 2025

The Impact of Hyperglycemia on Prognosis in Chronic Obstructive Pulmonary Disease Patients with Coronary Heart Disease: A Five-Year Prospective Study

PONE-D-25-26916R2

Dear Dr. Chen,

We’re pleased to inform you that your manuscript has been judged scientifically suitable for publication and will be formally accepted for publication once it meets all outstanding technical requirements.

Kind regards,

Hidetaka Hamasaki

Academic Editor

PLOS ONE

Additional Editor Comments (optional):

Reviewers' comments:

Reviewer's Responses to Questions

**Comments to the Author**

Reviewer #3: All comments have been addressed

2. Is the manuscript technically sound, and do the data support the conclusions?

Reviewer #3: Yes

3. Has the statistical analysis been performed appropriately and rigorously?

Reviewer #3: Yes

4. Have the authors made all data underlying the findings in their manuscript fully available?

Reviewer #3: Yes

5. Is the manuscript presented in an intelligible fashion and written in standard English?

Reviewer #3: Yes

Reviewer #3: Dear Authors,

It is a satisfaction to re-review this manuscript and observe that the authors addressed all the observations of the last review, adding more information to bring clarifications that improved the quality of the study.

**Do you want your identity to be public for this peer review?** For information about this choice, including consent withdrawal, please see our Privacy Policy

Reviewer #3: No

---

## [Editor Report · Acceptance letter]

PONE-D-25-26916R2

PLOS ONE

Dear Dr. Chen,

I'm pleased to inform you that your manuscript has been deemed suitable for publication in PLOS ONE. Congratulations! Your manuscript is now being handed over to our production team.

Kind regards,

on behalf of

Dr. Hidetaka Hamasaki

Academic Editor

PLOS ONE